# Efficiency of Tool Steel Cutting by Water Jet with Recycled Abrasive Materials

**DOI:** 10.3390/ma15113978

**Published:** 2022-06-02

**Authors:** Andrzej Perec, Aleksandra Radomska-Zalas, Anna Fajdek-Bieda, Elzbieta Kawecka

**Affiliations:** Faculty of Technology, Jacob of Paradies University, 66-400 Gorzów Wielkopolski, Poland; aradomska-zalas@ajp.edu.pl (A.R.-Z.); abieda@ajp.edu.pl (A.F.-B.); ekawecka@ajp.edu.pl (E.K.)

**Keywords:** abrasive water jet, AWJ, cutting depth, process efficiency, molybdenum tool steel, response surface method, RSM, modeling, optimization

## Abstract

High-pressure water jet machining is characterized by wide possibilities of cutting diverse materials together with multi-layer materials with dissimilar properties, accurate cutting complex profiles, as well as conducting them in uncommon conditions, especially in cases of thick materials. An additional advantage of water jet technology is its environmental friendliness. This paper presents tests of the cutting performance of tool steel with the use of an abrasive water jet (AWJ). The state-of-the-art has shown insufficient scientific evidence in AWJ tool steels cutting using recycled abrasive materials. Therefore, the main motivation for this paper was to carry out research from an environment aspect. The reuse of abrasives and the use of recycled materials have immense potential to reduce processing costs while remaining environmentally friendly. The RSM method was used for modeling and optimization. A response surface design (RSM) is a package of an advanced design-of-experiment (DOE) approaches that support better understanding and optimize response, exploring the relationships between several explanatory variables and one or more response variables. Based on this research, feed rate is the key factor influencing the depth of cut, while the water nozzle diameter has a secondary effect, and the concentration of abrasive has the least influence on the depth of cut. High level of variance (the percentage of variability in the reaction that is interpreted by the formula) confirms that the models fit well to the investigational data.

## 1. Introduction

High-pressure water jet treatment is one of the fastest-growing advanced production technologies. It effectively competes with conventional methods of materials separation. This is due to the versatile possibilities of cutting both a variety of materials [1] with different, even opposing properties [2] and accurate cutting complicated contours, or doing them in a special surround [3] (probability of detonation, fire, etc.).

The usual cutting methods for heavy- to machining metals do not provide adequate effectiveness and accuracy, especially in cases of thick materials. Abrasive water jet (AWJ) and pulsating water jet (PWJ) treatment [4,5] do not have these weaknesses. An additional benefit of WJ technology is its environmental friendliness because among the methods of machining only plastic processing [6,7] and AWJ treatment [8] can meet these requirements. 

Abrasive water jet (AWJ) is usually used in the treatment of heavy- to machining materials by conventional cut technologies, such as rocks [9], ceramics [10], composites [11], hard metals [12], superalloys [13], and even human bone [14].

Water under high pressure is transformed to a high-velocity jet, mixed with abrasive particles, and then formed in the working nozzle. The material removal occurs due to the combined effects of micro plastic deformation, micro cutting action of the high-velocity abrasive grits, and crack initiation and propagation.

The properties of abrasive grains [15,16], especially geometric parameters, are important in AWJ processes for achieving high efficiency and accuracy [17] on the treated surfaces.

Mahalingam et al. [18] suggested finding the optimum level of the chosen control parameters of AWJ, such as water pressure, stand-off distance, and abrasive flow rate, in the D2 steel drilling process under the contemporary minimization of the chosen drilled hole properties, such as surface and shape errors, by using the harmony search algorithm (HSA): the new metaheuristic approach.

Yuvaraj et al. [19], presented experiments on abrasive water jet (AWJ) cutting of D2 tool steel. The impact of the cutting control parameters as the water pressure, jet attack angle, and abrasive particle size on the outputs as the jet penetration, MRR, taper ratio, roughness, and topography, was tested. The total AWJ cutting efficiency of the D2 steel was increased through appropriate detection of the optimal control parameter level, by the Simos–Grey relational analysis. Additionally, the authors analyzed detailed 3D surface structure and 2D roughness contours were analyzed [20]. 

Kant et al. [21] presented the cutting of EN31 steel with an AWJ. In this research, pressure, abrasive flow rate, stand-off distance, and feed rate were selected as control parameters were studied for their impact on the output parameters treatment time and surface roughness. Based on Grey Relational Analysis, all the designated parameters were optimized for minimal treatment time and minimal surface roughness.

Pal et al. [22] tested the milling of chosen hard to machining materials by abrasive water jet. Results of the milling depth and material properties on machining time were studied. It was noted that machinability level and the mechanical properties of the materials cut played a key role in establishing cutting time and surface roughness. Among others, the tool steel M2 was used in the research. It was found that the traverse speed of the AWJ process is inferior for materials with a low machinability index, and contrarywise. Furthermore, the machining time increases non-linearly as the depth of cutting increases due to the energy loss of a jet and increase in stand-off distance.

Akkurt [23,24] published research of AWJ machining of different property metals including D3 cold working tool steel. The impact of thickness and material kind on the machining time were studied and considered. Outcomes of the hardness research demonstrated that there is not much relation between the drilled surface and the material hardness. Machining does not impact the microstructure and mechanical properties of the material.

Doreswamy et al. [25] presented the machining of high-carbon and high-chromium D2 steel using abrasive water jet (AWJ) to find the impact of stand of distance, the feed rate on the slot width, and on the Ra roughness parameter of the cut slot generated by AWJ. The results also showed that the increase in standoff distance and feed rate increases the surface roughness (Ra) value.

Hlavac performed analytical models [26] used for the explanation and estimation of AWJ machining of different metals, including the 1.7131 tool steel. The model showed AWJ’s potential to be a major tool for the preparation of steering software with higher accuracy in determination of effects and higher computation rate.

The phenomenon of disintegration of abrasive grains in the subsequent stages of high-pressure abrasive water jet creation and the same material treatment of this abrasive water jet was presented by Prazmo et al. [27]. They investigated garnet breakage intensity in AWJ cutting during the generation of the AWJ and in the whole cutting operation. An analysis of the abrasive recovered after treatment was carried out and the abrasive erosion efficiency of the abrasive was evaluated. The authors also extensively analyzed the possibility of abrasive material re-use and economic aspects of this operation.

Whereas Gent et al. [28,29] introduced research on the effects of a few mineral abrasive and one high-density glass abrasive to recognize the abrasive properties suitable for the machining of ductile materials by AWJ. The authors showed that above a certain abrasive density there is no improvement in rate of this type of erosion and that polycrystalline abrasives are more effective than mono-crystalline abrasives of the same composition.

Material machining by abrasive water jet is more complicated in relation to conventional treatments and therefore modeling and optimization [30,31,32] were used to achieve good machining results. Modeling and optimization by various methods were used in other fields, for example, both in surface machining [33,34], catalytic systems [35,36], epoxidation processes [37,38], research into human bone [39], and even in digital signal processing [40,41] and positioning systems [42].

The analysis of the state of the problem showed that the processing of materials from the group of tool steels is possible by AWJ, but until now, no recycled abrasives have been used. To date, research with such materials has been published only in the areas of drilling and comparison of erosion rates. Therefore, this paper presents new research on recycled materials for cutting through hard-to-machine tool steel, and in this area the paper presents scientific novelty. This is an issue worthy of attention as it will allow to eliminate large amounts of used abrasive for the benefit of its reuse.

## 2. Materials and Methods

### 2.1. Cut Material

Molybdenum high-speed steels and tungsten high-speed steels belong to one tool-steel group. Molybdenum high-speed tool steels have less initial cost under similar efficiency performance. High-speed steel tools can be coated with titanium carbide, titanium nitride, and other coatings by direct deposition under a vacuum process for extending performance and prolonging the life of the tool. There are several types of molybdenum high speed tool steels. Type AISI M1 steel (DIN 1.3346 and SAE J438 equivalent), which was chosen in research consisting of some tungsten, but no cobalt. The detail of the chemical composition of M1 molybdenum high speed tool steels is tabulated in Table 1 and its physical and mechanical properties in Table 2.

Thanks to the unique properties of this steel, it is appropriate for making all kinds of cutting tools.

### 2.2. Abrasive Materials

One of the abrasives used in the research is an iron–aluminum garnet–almandine. Magnesium can replace iron and become more like pyrophoric magnesium aluminum garnet. Pure almandine and pure pyrope are rare in nature, and most of the minerals are a mixture of these two. Almandine, like other garnet, forms isometric crystals with 12 orthorhombic or 24 trapezoidal surfaces, or combinations of these and other forms. Shapes of real grains are shown in Figure 1a.

The GMR80 is a GMA alluvial almandine garnet recycling product, manufactured by GMA Garnet (Middle East) FZE in Jebel Ali Dubai, United Arab Emirates. The GMA Garnet abrasive can be recycled up to five times without any loss in performance [44]. Improvements in cutting effectiveness are achievable when using recycled abrasive [27]. This is due to its fair erosion proof and low fragility.

The grain size distribution of the recycled GMR80 garnet used in the research is shown in Figure 1a. Density function of distribution is near symmetric with a predominance of grains of 180 μm and 150 μm size. 

Properties of GMR80 garnet are specified in Table 3. The most common glass abrasive is the product produced by crushing sodium–calcium glass. It is gained from fragmented and divided products of cullet glass, or recycled materials of industrial products. 

The use of natrium–calcium glass-based abrasives in AWJ parting and AWJ surface treatment is not usual. They are applied only as experimental abrasives. The tests were carried out with small glass granules.

The Na-Ca glass-based abrasive is of inferior performance when cutting steel and raw rock due to its high brittleness. This abrasive is only recommended for cutting soft materials: wood, plastic, and rubber. The dust resulting from fragmentation during treatment is neutral to the environment [45]. The abrasive obtained from high-density industrial glass (SPDG60) brings with it greater prospects of application. This abrasive was also used in our tests (Table 3). In the tested material, negative asymmetry with a predominant grain size from 250 µm to 212 µm (Figure 1b) was observed.

Recycling provides abrasive users with a cost-effective and ecologically sound opportunity to dispose of used abrasive that would otherwise be collected as industrial waste. Recycling also increases the service life of this non-renewable natural raw material and significantly reduces the volume of abrasive used [46].

### 2.3. Response Surface Methodology (RSM)

A response surface design is a package of advanced design of experiments (DOE) approaches that support better understanding and optimize response. Response surface methodology (RSM) design is often used to refine models after key factors were determined by using screening designs or factorial designs, especially if curvature in the response surface is considered. RSM explores the relationships between several explanatory variables and one or more response variables. The primary idea of RSM is to use a sequence of designed experiments to obtain an optimal response. 

In the research, RSM was used to fit a complete quadratic polynomial model through central composite experiment, and it achieved more excellent experiment design and result expression.

### 2.4. Test Set up and Test Method

The experiments were carried out on a test stand built based on the high-pressure intensifier type I50 by KMT, CNC device type ILS55 by Techni Waterjet and dedicated steering system. The maximum available constant pressure is over 400 MPa at water flow near 5 dm^3^/min. The maximum diameter of a water nozzle at this condition is 0.4 mm. The abrasive quantity was dosed by manually change the abrasive inlet area. The values of the control parameters characterizing the AWJ cutting process were determined based on our previous works [47], and the works of other researchers: Alberdi et al. [48], Lehocka et al. [49], and Pude et al. [50]. The details of cutting parameters and their levels are presented in Table 4.

The impact of the tested control factors on the process efficiency were presented in Table 4. The other parameters, such as pressure = 390 MPa and stand-off distance = 4 mm, remained constants during the research. The cutting process at each set of control parameters was repeated 3 times.

## 3. Results and Discussion

### 3.1. Results of Research

The control factors impact on the process efficiency was carried out using analysis of variance (ANOVA). Grounded on the effects showed in Table 5, the importance of treatment parameters diameter of water nozzle, concentration of abrasive, and feed rate for both abrasives on cutting depth were calculated.

ANOVA was performed for a 95% confidence interval (α = 0.05). The *p*-value < 0.05 entails that the model coefficient is meaning.

This is also shown in detail in Figure 2. The factors to the left of the black dashed line are statistically insignificant (at 0.95% confidence level) and in Equations (1) and (2) their effect on cut depth was neglected. The factors that pass the test of significance are considered significant. They are considered insignificant if they fail the test of significance and are usually treated as if they are not present. This process is called pooling.

The outcomes of the ANOVA are presented in Table 6 for GMR80 abrasive and in Table 7 for SPDG60 abrasive. 

Feed rate is the most significant factor influencing the assessment for cutting depth. Meanwhile, water nozzle ID has a less significant effect on cutting depth. Abrasive flow rate has the least significant impact on cutting depth. The influences of the above control parameters were similar for both tested abrasives. 

On ground of coefficients (Table 6 and Table 7), the final cutting depth control models were specified:(1)$$h_{\left(GMR80\right)}=7.7-152d_{w}+3.34c_{a}-3.48v_{p}+444d_{w}^{2}-0.09c_{a}^{2}+0.62v_{p}^{2}-15.56d_{w}v_{p}$$
(2)$$h_{\left(SPDG60\right)}=1.3-100.6d_{w}+3.43c_{a}-5.57v_{p}+304d_{w}^{2}-0.09c_{a}^{2}+0.68v_{p}^{2}-8.17d_{w}v_{p}$$
where:

*h_(GMR80)_* is cutting depth achieved with GMR80 abrasive material,

*h_(SPDG60)_* is cutting depth achieved with SPDG60 abrasive material,

*d_w_* is water nozzle diameter [mm],

*c_a_* is concentration of abrasive [%],

*υ_p_* is feed rate [mm/s]

These equations are illustrated in Figure 3 (Equation (1)). The dependence of the depth of the cut on the feed is inversely proportional. Increasing the feed leads to a reduction in the depth of cut over the entire tested range. The biggest cutting depth was achieved with the largest nozzle, with the lowest feed and concentration of just over 20.5%.

The graphs of the dependence of the cut depth on the inner diameter of the water nozzle showed that the depth of cut is directly proportional to the diameter of the nozzle and reaches the highest values for the largest nozzle diameter. The diameter of the water nozzle had a smaller influence on the achieved cutting depth than the feed. The biggest depth of cut was caused by the biggest water nozzle at the smallest feed.

The influence of abrasive concentration on the cutting depth was the smallest. The maximum cutting depth could be observed for the abrasive concentration of 20% at the lowest feed and the largest diameter of the water nozzle. Both reducing and increasing the concentration of abrasive in the stream resulted in a decrease in the cutting depth.

What is more, when using abrasive SPDG60, dependence of the depth of cut on the feed rate, described Equation (2) and illustrated in Figure 4, was inversely proportional. Increasing the feed led to a reduction in the depth of cut over the entire tested range. The graphs of the dependence of the cut depth on the inner diameter of the water nozzle show that the depth of the cut is directly proportional to the diameter of the nozzle and reaches the highest values for the largest nozzle diameter. The diameter of the water nozzle had a smaller influence on the achieved cutting depth than the feed. The greatest depth of the cut was achieved with the nozzle with the largest diameter and lowest feed. The influence of abrasive concentration on the cutting depth was the smallest. The maximum cutting depth could be observed with a concentration of abrasive of 19.9%. Both reducing and increasing the concentration of abrasive in the jet resulted in a decrease in the cutting depth.

Based on the analysis of variance (Table 7), it can be observed that the feed is the machining parameter that most influences the depth of cut. 

Low feed (2 mm/s) allowed cutting by more abrasive grains. The high number of abrasive grains provided multiple cutting edges which led to an increased material removal rate. In this way, a deeper cut was achieved. The value of the most effective mass flow of grains in the stream (concentration) was almost 20%.

Beyond this level, the water jet was unable to supply the maximal speed for all abrasive particles. Another unfavorable element was the collisions between the abrasive particles, which also destructively affected the overall speed (energy) of the AWJ.

Reducing the concentration below 20% reduced the depth of cut due to fewer abrasive grains and therefore fewer cutting edges. The character of this dependence was observed in cases of all nozzles using all tested abrasive materials.

The compatibility of the models was checked using the function R^2^. In regression, the coefficient of determination R^2^ is a statistical measure of how closely the regression line approximates the actual data points. Table 8 shows the regression standard error, R^2^, R^2^ adjusted, and R^2^ predicted. 

The values of all the coefficients of determination R^2^ are greater than 98%. High and only slightly different values demonstrated by the raw data fit very well with the regression line. The scatter plot (Figure 5) confirms this compliance.

It can be seen that the points are close to the straight cyan line. This suggests that the developed mathematical models of the tool steel cutting process for both GMR80 and SPDG60 abrasives are satisfactory.

Details of calculated influence of the control parameters on the maximum cutting depth are shown in Figure 6.

For GMR80 abrasive, the maximal depth of cut was reached at the concentration of 20.5% and amounted to 28.39 mm. For the SPDG60 abrasive, the maximum cutting depth was 21.98 mm, and it was achieved with a concentration of 19.93%. For both abrasives, the maximal depth of cut was reached for the major water nozzle ID and the lowest feed rate.

### 3.2. Results Discussion

Samples of SEM cut surface photos achieved under optimal control parameters are presented Figure 7. Shallow machining marks deepening in the lower part of the material were observed. Under 200× magnification parallel traces of micro-cutting were visible in the upper part (I) of the sample, became less ordered in the middle (II), and finally in the lower zone (III). Particularly, the characteristic signs of erosion by the abrasive grains could be observed in Figure 7c,e,g. They are visible in the form of parallel tracks.

Abrasive grains, trapped in the workpiece, were also observed on the cut surface (Figure 8). This was confirmed by the EDS analysis, as both oxygen, silicon, and aluminum atoms were detected in the grain. However, their occurrence was sporadic. A similar shape was also taken by the chips formed during the cutting of the material.

In comparison to the new garnet abrasive—GMA80 [51], we can observe even 1.7% higher average cutting depth (Figure 9) when using recycled abrasive (GMR80). For crushed high density glass abrasive (SPDG60) the average cutting depth is 20% lower. However, the low cost of this recycled abrasive may make the use of this abrasive an economically viable alternative.

The results obtained from the tests are generally in line with the results of other studies. In particular, in terms of the impact of feed rate [21,22] or the flow of abrasive [18,25]. This confirms the earlier observations [51] that the nature of the impact of the key control parameters on the AWJ cutting process is similar for diametrically different abrasives. 

The main difference remains in the efficiency, most often referred to as the maximum depth of cut [19]. The research also confirmed the higher efficiency of machining tool steel by AWJ with the use of recycled abrasive, similarly to the effects on cutting aluminum [27].

## 4. Conclusions

The scientific novelty brought by this work is related to the study of the possibility for cutting difficult-to-machine tool steel by AWJ with recycled abrasive material. Based on the studies, the legitimacy of using recycled abrasive was shown; it resulted in filling a gap in the literature and the following conclusions were reached:All the analyzed AWJ control factors have a big effect on erosive ability, as determined by depth of cut.The depth of the cut is directly proportional to the diameter of the water nozzle in the whole tested range.R^2^ (the percentage of response variability that is explained by the model) near 99% demonstrates both models fit very nicely to investigational data.Adjusted R^2^ amount over 98.5% also validates an exceptionally good model fit.The predicted R^2^ above 98% demonstrates particularly good prediction of the model response to new observations.For the model regression factors no multicollinearity was notedThe feed rate is the key factor affecting the depth of the cut. The water nozzle diameter has a secondary effect, and the concentration of abrasive has the least influence on the depth of the cut.The optimum values of the AWJ control parameters from the cutting depth are as follows: water nozzle diameter = 0.33 mm, feed rate = 2 mm/s, GMR80 concentration = 19.93%, and SPDG60 concentration = 20.53%. At these control parameter values, the highest depth of cut for the GMR80 abrasive was 28.39 mm and for the SPDG60 abrasive was 21.98 mm.

At this stage, the work is purely scientific. In the further research, the cutting models will be expanded with additional control parameters, such as abrasive size or focusing tube length. Additionally industrial application potential, especially in the conditions of increasing abrasive prices, seems to be high.

## Figures and Tables

**Figure 1 materials-15-03978-f001:**
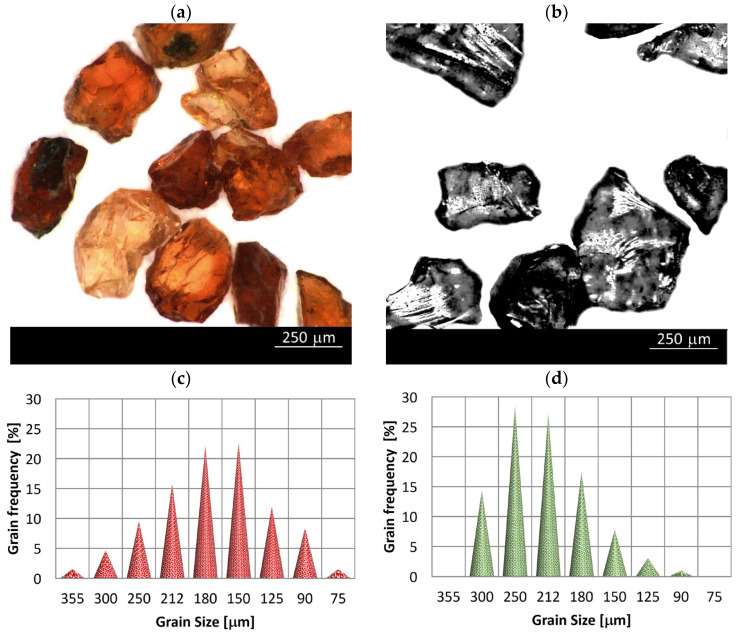
Grain view of: (**a**) GMR80 recycled garnet abrasive; (**b**) SPDG60 recycled glass abrasive and grain size distribution of: (**c**) GMR80 recycled garnet abrasive; (**d**) SPDG60 recycled glass abrasive.

**Figure 2 materials-15-03978-f002:**
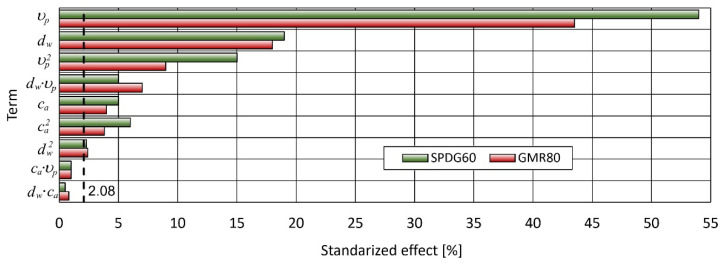
Pareto chart of the cutting standardized effects for GMR80 recycled garnet abrasive and SPDG60 glass abrasive.

**Figure 3 materials-15-03978-f003:**
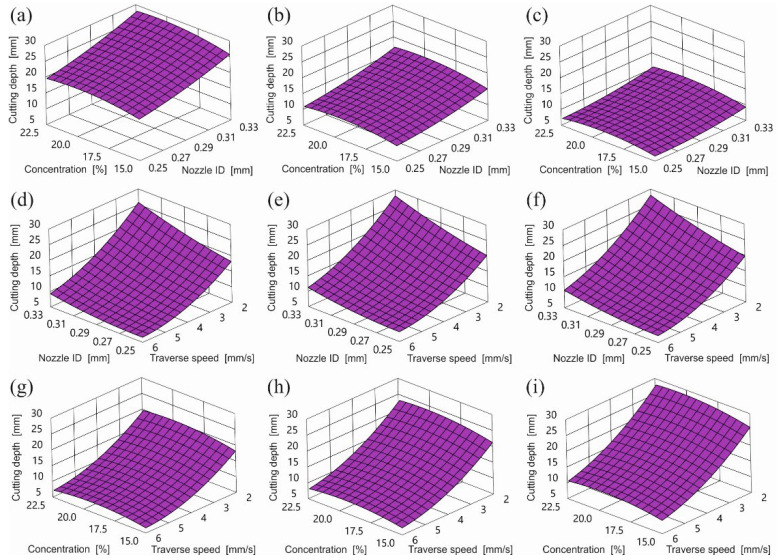
Influence of control parameters on cutting depth for GMR80 recycled garnet abrasive in conditions: (**a**) feed rate 2 mm/s, (**b**) feed rate 4 mm/s, (**c**) feed rate 6 mm/s, (**d**) abrasive concentration 15%, (**e**) abrasive concentration 18.7%, (**f**) abrasive concentration 22.5%, (**g**) water nozzle ID 0.25 mm, (**h**) water nozzle ID 0.29 mm, (**i**) water nozzle ID 0.33 mm.

**Figure 4 materials-15-03978-f004:**
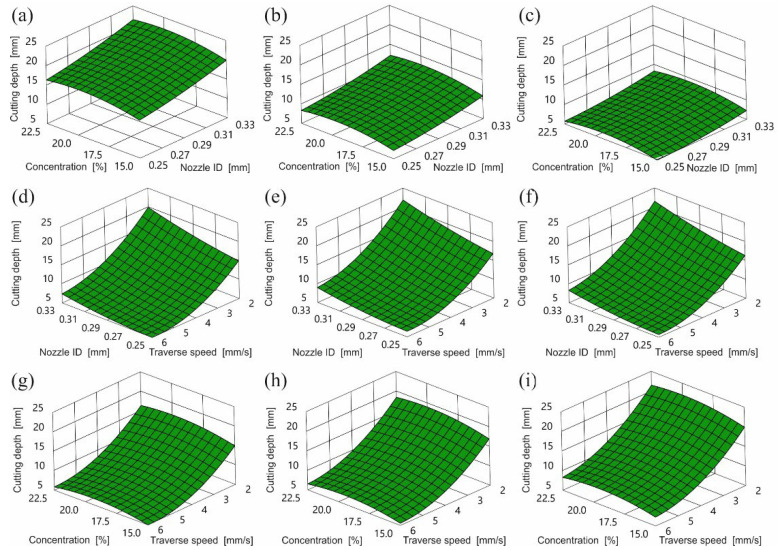
Influence of control parameters on cutting depth for SPDG60 recycled garnet abrasive in conditions: (**a**) feed rate 2 mm/s, (**b**) feed rate 4 mm/s, (**c**) feed rate 6 mm/s, (**d**) abrasive concentration 15%, (**e**) abrasive concentration 18.7%, (**f**) abrasive concentration 22.5%, (**g**) water nozzle ID 0.25 mm, (**h**) water nozzle ID 0.29 mm, (**i**) water nozzle ID 0.33 mm.

**Figure 5 materials-15-03978-f005:**
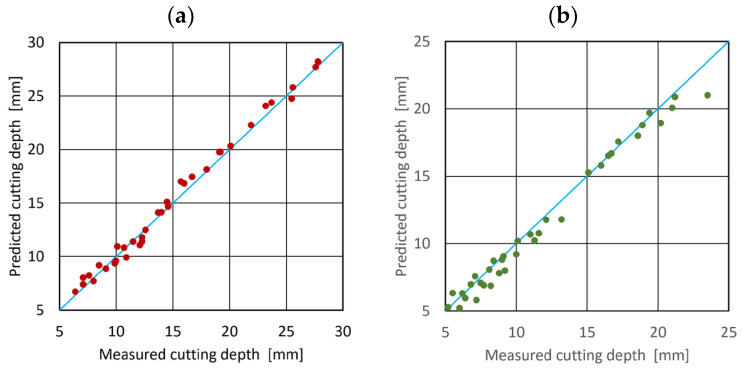
Scatter chart of the model and measured depth of cut for (**a**) GMR80 recycled garnet abrasive material, (**b**) SPDG60 glass abrasive material.

**Figure 6 materials-15-03978-f006:**
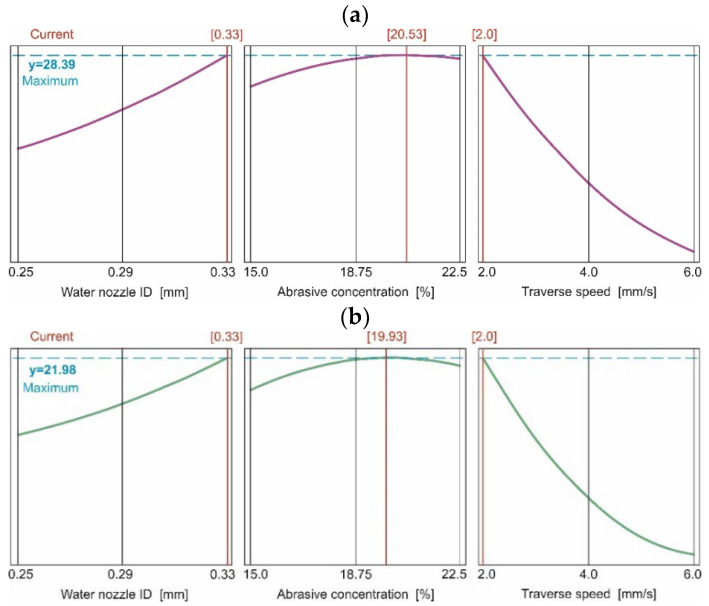
Chart of the process parameters input on cutting depth for (**a**) GMR80 recycled garnet abrasive material, (**b**) SPDG60 glass abrasive material.

**Figure 7 materials-15-03978-f007:**
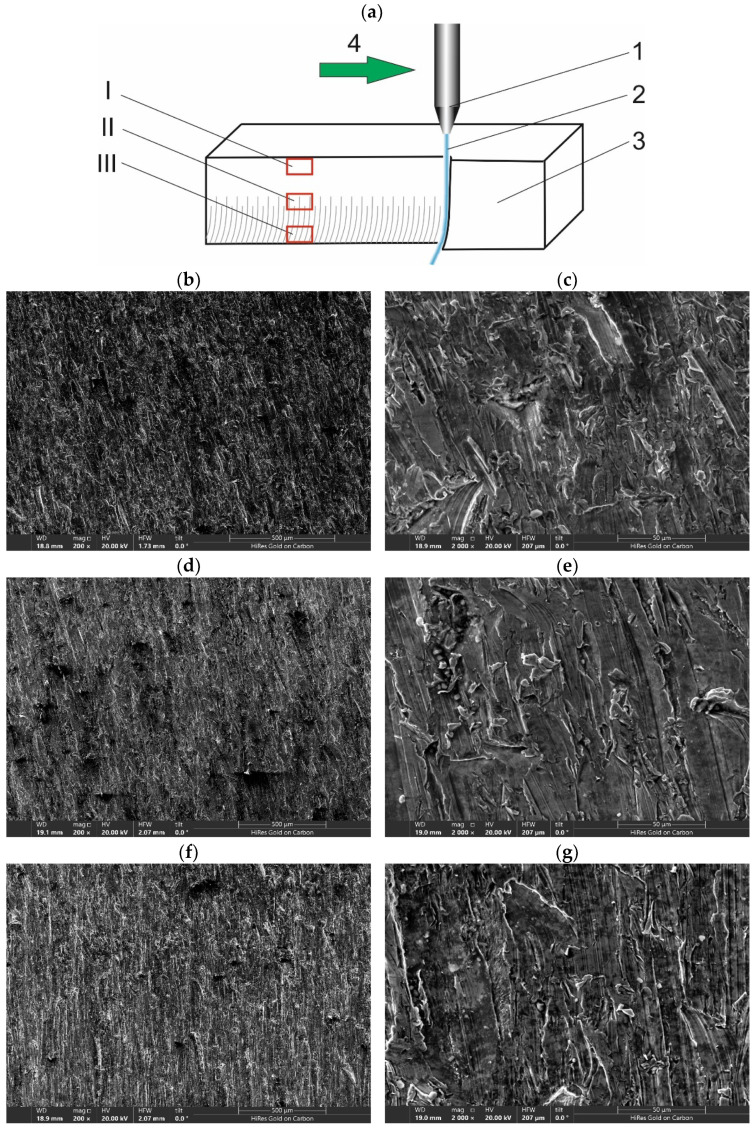
Sample SEM microscope view of cut surface (**a**) details of cut surface area location: 1-working nozzle, 2-AWJ, 3-target, 4-nozzle move direction, I-top zone, II-middle zone, III-bottom zone, (**b**) top surface, magn. 200×, (**c**) top surface, magn. 2000×, (**d**) middle surface, magn. 200×, (**e**) middle surface, magn. 2000×, (**f**) bottom surface, magn. 200×, (**g**) bottom surface, magn. 2000×.

**Figure 8 materials-15-03978-f008:**
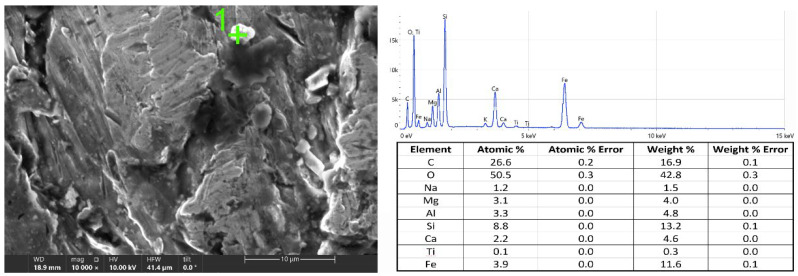
Sample of abrasive grain in the cut surface.

**Figure 9 materials-15-03978-f009:**
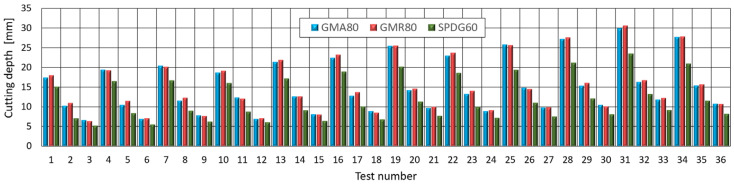
Comparison of cutting depth achieved by recycled abrasive (GMR80 and SPDG60) with non-used abrasive (GMA80).

**Table 1 materials-15-03978-t001:** Chemical composition of M1 tool steel [43].

Elment	C	Mn	Si	Cr	Ni	Mo	W	V	Cu
Content [%]	0.78–0.88	0.15–0.40	0.2–0.5	3.5–4.0	0.3	8.2–9.2	1.4–2.1	1.00–1.35	0.25

**Table 2 materials-15-03978-t002:** Physical and mechanical properties of M1 tool steel.

Physical Properties		Mechanical Properties	
Density [kg/m^3^]	7890	Hardness, Rockwell C	60–65
Yield Tensile Strength [MPa]	417	Poisson’s ratio	0.27–0.30
Ultimate Tensile Strength [MPa]	712	Elastic modulus [GPa]	190–210

**Table 3 materials-15-03978-t003:** Recycled garnet and high-density industrial glass abrasive properties.

	GMR80	SPDH60
**Crystallographic system**	Cubic	Amorphous
**Mohs hardness**	6.5–7.5	5.0
**Density [kg/m^3^]**	4100–4300	3700
**Fracture**	Shell to uneven	Shell, very sharp, sword-like grains
**Color**	From intensive red to red brown, occasionally with a brown or purple tinge, to dark brown	Brown-black, gray-black, black

**Table 4 materials-15-03978-t004:** Cutting parameters.

Symbol	Factor	Level			
		1	2	3	4
*d_w_*	Diameter of water nozzle [mm]	0.25	0.3	0.33	-
*A_c_*	Concentration of abrasive [%]	15.0	17.5	20.0	22.5
*υ_p_*	Feed rate [mm/s]	2	4	6	-
*T_a_*	Abrasive type	GMR80	SPDG60	-	-

**Table 5 materials-15-03978-t005:** Controls and dependent parameters of AWJ cutting.

Test Nr.	Diameter of Water Nozzle	Concentration of Abrasive	Feed Rate	Depth of Cut for:	
				GMR80	SPDG60
	[mm]	[%]	[mm/s]	[mm]	[mm]
1	0.25	15.0	2	18.0	15.1
2	0.25	15.0	4	10.9	7.1
3	0.25	15.0	6	6.4	5.2
4	0.25	17.5	2	19.2	16.5
5	0.25	17.5	4	11.5	8.4
6	0.25	17.5	6	7.1	5.5
7	0.25	20.0	2	20.1	16.7
8	0.25	20.0	4	12.3	9.0
9	0.25	20.0	6	7.6	6.2
10	0.25	22.5	2	19.1	16.0
11	0.25	22.5	4	12.1	8.8
12	0.25	22.5	6	7.1	6.0
13	0.30	15.0	2	21.9	17.2
14	0.30	15.0	4	12.6	9.1
15	0.30	15.0	6	8.0	6.4
16	0.30	17.5	2	23.2	18.9
17	0.30	17.5	4	13.7	10.1
18	0.30	17.5	6	8.5	6.8
19	0.30	20.0	2	25.5	20.2
20	0.30	20.0	4	14.6	11.3
21	0.30	20.0	6	10.0	7.7
22	0.30	22.5	2	23.7	18.6
23	0.30	22.5	4	14.0	10.0
24	0.30	22.5	6	9.1	7.2
25	0.33	15.0	2	25.6	19.4
26	0.33	15.0	4	14.5	11.0
27	0.33	15.0	6	9.9	7.5
28	0.33	17.5	2	27.6	21.2
29	0.33	17.5	4	16.0	12.1
30	0.33	17.5	6	10.1	8.1
31	0.33	20.0	2	29.9	23.5
32	0.33	20.0	4	16.7	13.2
33	0.33	20.0	6	12.3	9.2
34	0.33	22.5	2	27.8	21.0
35	0.33	22.5	4	15.7	11.6
36	0.33	22.5	6	10.7	8.2

**Table 6 materials-15-03978-t006:** Analysis of variance parameters of AWJ cutting with VIF factors for GMR80 abrasive.

Source	DF	Adj SS	Contribution	Adj MS	F-Value	*p*-Value	VIF
Model	9	1565.97	98.95%	174.00	273.07	0.000	
Linear	3	1429.52	93.21%	476.51	747.83	0.000	
*d_w_*	1	181.72	11.29%	181.72	285.19	0.000	1.02
*c_a_*	1	10.82	0.79%	10.82	16.98	0.000	1.01
*υ_p_*	1	1236.90	81.13%	1236.90	1941.19	0.000	1.01
Square	3	63.48	4.03%	21.16	33.21	0.000	
*d_w_* ^2^	1	3.48	0.22%	3.48	5.46	0.027	1.02
*c_a_* ^2^	1	10.25	0.67%	10.25	16.09	0.000	1.00
*υ_p_* ^2^	1	49.75	3.15%	49.75	78.08	0.000	1.00
2-Way Interaction	3	26.60	1.70%	8.87	13.91	0.000	
*d_w_·c_a_*	1	0.55	0.03%	0.55	0.86	0.363	1.01
*d_w_ ·υ_p_*	1	25.30	1.62%	25.30	39.71	0.000	1.01
*c_a_ ·υ_p_*	1	0.75	0.05%	0.75	1.18	0.288	1.00
Error	26	16.57	1.05%	0.64			
Total	35	1582.53	100.00%				

SS sum of squares, DF degree of freedom, MS mean square, F ratio of variance of a source to variance of error, VIF variance inflation factor.

**Table 7 materials-15-03978-t007:** Analysis of variance parameters of AWJ cutting with VIF factors for SPDG60 abrasive.

Source	DF	Adj SS	Contribution	Adj MS	F-Value	*p*-Value	VIF
Model	9	991.589	99.33%	110.177	418.64	0.000	
Linear	3	891.532	91.54%	297.177	1129.19	0.000	
*d_w_*	1	85.744	8.49%	85.744	325.80	0.000	1.02
*c_a_*	1	7.103	0.79%	7.103	26.99	0.000	1.01
*υ_p_*	1	798.640	82.26%	798.640	3034.61	0.000	1.01
Square	3	70.260	7.06%	23.420	88.99	0.000	
*d_w_* ^2^	1	1.628	0.16%	1.628	6.19	0.020	1.02
*c_a_* ^2^	1	9.951	1.03%	9.951	37.81	0.000	1.00
*υ_p_* ^2^	1	58.681	5.87%	58.681	222.97	0.000	1.00
2-Way Interaction	3	7.283	0.74%	2.428	9.22	0.000	
*d_w_·c_a_*	1	0.015	0.00%	0.015	0.06	0.810	1.01
*d_w_ ·υ_p_*	1	6.979	0.70%	6.979	26.52	0.000	1.01
*c_a_ ·υ_p_*	1	0.288	0.03%	0.288	1.10	0.305	1.00
Error	26	6.843	0.67%	0.263			
Total	35	998.432	100.00%				

SS sum of squares, DF degree of freedom, MS mean square, F ratio of variance of a source to variance of error, VIF variance inflation factor.

**Table 8 materials-15-03978-t008:** Analysis of variance parameters of AWJ cutting.

Abrasive	S	R^2^ [%]	R^2^_(adj)_ [%]	R^2^_(pred)_ [%]
GMR80	0.801	98.95	98.58	98.06
SPDG60	0.508	99.33	99.09	98.71

## Data Availability

Not applicable.

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
