# Peer review of "Efficiency of Tool Steel Cutting by Water Jet with Recycled Abrasive Materials"

_materials, 2022, doi:10.3390/ma15113978_

Round 1

Reviewer 1 Report

Good circular economy application, the use of old abrasives can be interesting in AWJ, as it was in some classic works. In ASME by Alberdi,  https://doi.org/10.1115/1.4004320 was a previous definition, and the idea was use in massive AWJ for milling, check it and complete. This is the right reference not the 47, or at least you can complete. ASME is very important. In milling the impact is the key parameter, in cutting perhaps the uniformity. Complete the view.

What is ref 42??

Figure 7 is very small

Table 1 is from the standard or was specific for your case??

ANOVa is well performed, however the focus must be the real manufacturing results.

In AWJ milling there are issues about grains removing (see above missed reference)…was there any instance of grain imbibed in your surfaces?.

What is your opinion of boosting ensembles, Ensemble learning is a machine learning paradigm where multiple models (often called “weak learners”) are trained to solve the same problem and combined to get better results…Bustillo for instance gave some improvements in friction processes.

Main conclusions: recycled grains can be a solution: have you a comparison of a new and second use grain sample?

Reviewer 2 Report

Dear Authors,

Congratulations on your work, which is focused on a very interesting subject. As any other paper in this phase, there are some amendments to do, whose can improve the overall quality of your paper. Thus, I'm providing below some comments and suggestions, trying to collaborate by this way in improving your paper:

  1. The Abstract doesn't clearly state the literature gap found, as well as the main motivation to develop this work. Thus, please clearly state the gap found in the literature in the Abstract, Introduction and Conclusions. The mains goals are also not clear in the Abstract..
  2. The novelty brought by your work is also not properly pointed out. Thus, please state clearly the novelty that your paper represents for the scientific community, stating as well if your contribution is exclusively scientific or if there was some practical motivation behind the development of your work. Any industrial application based on this work should also be pointed out.
  3. Keywords should include the methods used to compute the results (statistics methods).
  4. The first 22 references are spent in generalities. Just reference [23] starts with direct speech about previous works carried out and corresponding results. Please improve this, reducing the references used regarding generalities, and improving the number of references used to describe works and results previously obtained by other Researchers. Moreover, please use these references to perform a deep DISCUSSION of your results, crossing information/results.
  5. In Table 2, please include the Ultimate and Yield Tensile Strength value for the alloy used.
  6. In the methodology, please refer the number of experiments made under each condition.
  7. Usually, the abrasive particles change their geometry under use. Moreover, abrasive particle fragmentation is often in this process. Please refer as you have controlled these effects along your tests.
  8. DISCUSSION abou RESULTS needs to be included. This is mandatory.
  9. Conclusions do not sound. Please try to correlate them with the novelty of your work, which needs to be deeply highlighted.

Best wishes.

Kind regards. 

Reviewer 3 Report

In this work, the authors used response surface design and DOE methods studying AWJ cutting of tool steel with recycled abrasives. Some experimental results were presented with corresponding explanations. Please see following the detailed comments and suggestions.
1. The writing of abstract is poor, please modify and rewrite it, in order to highlight the key findings and objectives of this work.
2. Please avoid listing the references/reserach conclusions in the introduction, it should clearly present the summary and research gap.
3. The workpiece material selected for the research was very common one in application, what is the main contribution or scientic reason for using this type of material?
4. RSM method is widely used for a few years, the detailed description of this method can be deleted in the second section.
5. For the ANOVA analysis, what are the persentatge of contributions of each factor? what is the error level?
6. In AWJ machining process, the hydraulic pressure and stand-off distance are significant factors, while they were not considered in the work. What are the main reasons?
7. Very limited cutting surface was presented in the manuscript, what are the differences amony other trials? The defects cannot be found in the picture with very low resolution.

Round 2

Reviewer 1 Report

Ok

Author Response

Thank You very much for Your thorough review, proofreading, and time spent commenting on our work.

Reviewer 2 Report

Dear Authors,

The paper is good but is too much supported by your previous work. Dr. A. Perec has 14 references in 49. The remaining Authors have other references. Are there other Authors working in the same field? Please recognize the work of other Colleagues out of Poland, who have published in this field. Thus, please reduce the number of self-citations.

Kind regards.

Author Response

Thank You very much for Your thorough review, proofreading, and time spent commenting on our work.
In our responses, we have explained the details of the manuscript revision and our responses to the reviewers' comments.

The number of self-citations has been reduced below 25%, a level usually accepted in scientific publications.
Citations from non-Polish authors have also been added.

Reviewer 3 Report

The revision is acceptable. Good luck.

Author Response

(The authors gave the same response as above.)
